# Verifying the accuracy of Japanese version of the pediatric delirium assessment scale: SOS-PD and the high accuracy of family assessments of pediatric delirium

Yujiro Matsuishi[1], Haruhiko Hoshino[2], Yuki Enomoto[3,4], Takahiro Kido[3,4], Asaki Matsuzaki[5], Nobutake Shimojo[3], Bryan J. Mathis[6], Joshua Gallagher[7], Erwin Ista[8,9], Yoshiaki Inoue[3]*

1 Adult and Elderly Nursing, Faculty of Nursing, Tokyo University of Information Science, Chiba, Japan, 2 Department of Nursing, Faculty of Medical Technology, Teikyo University, Tokyo, Japan, 3 Department of Emergency and Critical Care Medicine, Institute of Medicine, University of Tsukuba, Tsukuba, Ibaraki, Japan, 4 Department of Pediatrics, University of Tsukuba Hospital, Tsukuba, Ibaraki, Japan, 5 Department of Psychiatry, Institute of Medicine, University of Tsukuba, Tsukuba, Ibaraki, Japan, 6 Department of Cardiovascular Surgery, Institute of Medicine, University of Tsukuba, Tsukuba, Ibaraki, Japan, 7 Department of Public Health and Primary Care, Cambridge Institute of Public Health, School of Clinical Medicine, University of Cambridge, Cambridge, United Kingdom, 8 Pediatric Intensive Care and Department of Pediatric Surgery, Erasmus University Medical Center - Sophia Children's Hospital, Rotterdam, The Netherlands, 9 Department of Internal Medicine, Section Nursing Science, Erasmus University Medical Center, Rotterdam, The Netherlands

* yinoue@md.tsukuba.ac.jp

## Abstract

### Background

Detecting pediatric delirium in critically ill children is important. The Sophia Observation withdrawal Symptoms and Delirium scale (SOS-PD) is a tool for assessing both pediatric delirium and iatrogenic withdrawal symptoms and contains a question for parents to assist in detecting pediatric delirium.

### Objectives

The aim was to translate the Japanese SOS-PD and to perform a cross-culture validation of the J-SOS-PD pediatric delirium dimension while confirming the accuracy of family assessments of pediatric delirium.

### Methods

The translation was undertaken with the internationally established forward- backward translation method. Pediatric delirium was simultaneously evaluated and compared between psychiatric diagnoses based on assessment by a pediatric intensivist and the Japanese version of the SOS-PD as evaluated by PICU researchers. We evaluated the criterion validity (sensitivity and specificity), cut-off point using a

**Data availability statement:** The datasets generated and analyzed during this study are not publicly available due to the presence of structured and sensitive information that, if openly accessible, could be misinterpreted or misused beyond the scope of the approved research protocol. However, in accordance with ethical research principles and transparency, the dataset is available upon reasonable request. Researchers interested in accessing the data for future studies may contact the corresponding author or the Tsukuba Clinical Research Ethics Committee, which is responsible for handling data access requests. Requests may be submitted to the committee at t-cred. adm@un.tsukuba.ac.jp and will be reviewed to ensure compliance with ethical and legal requirements before access is granted.

**Funding:** This work was supported by the Grant-in-Aid for Scientific Research in Japan (22K17480).

receiver operating characteristic (ROC) curve, and reliability with Cohen's κ coefficient and intraclass correlation coefficients (ICC).

## Results

A total of 125 independent assessments were performed in 67 children with a median age of 15 (IQR 5, 54) months and with a pediatric delirium-positive rate of 30% based on psychiatric evaluation. Based on the ROC curve analysis, the cut-off point of 4 was the most appropriate within the original scale and the Japanese SOS-PD version showed high sensitivity (0.92, 95% CI 0.84–1.00) and specificity (0.97, 95% CI 0.94–1.00) at a cut-off point of 4 and high reliability within the researcher assessments (κ = 0.95). We also verified family assessments of pediatric delirium as showing high sensitivity (0.90) and specificity (0.89) over 36 assessments.

## Conclusions

The Japanese version of the SOS-PD shows a high accuracy similar to the original. Moreover, we revealed high accuracy in family perception of pediatric delirium that could promote family presence in PICU settings.

## Introduction

Delirium features disturbances in attention, cognitive failures that occur within a short period of time, and additional cognitive impairments [1]. Pediatric delirium, in particular, has been well investigated within the literature and its prevalence is reported to be from 20–44% [1,2]. It is associated with higher mortality [3], longer length of hospital stay, and declines in cognitive function from intensive care unit (ICU) admission to discharge [4], making it a serious healthcare issue [5]. In consideration of its harmful effects, early detection for pediatric delirium is critical to mitigate adverse events. However, there are many symptoms, including pain, sedation, and iatrogenic withdrawal syndrome [6], that might be confused with symptoms of pediatric delirium. To this end, several validated assessment tools, including the Sophia Observation withdrawal Symptoms and Delirium (SOS-PD) scale, have been developed [7]. The SOS-PD consists of 22 symptoms with overlapping indications, 5 of which are only for iatrogenic withdrawal symptoms while 7 are only for pediatric delirium, and 10 are for both iatrogenic withdrawal symptoms and pediatric delirium, making it helpful for clinical practice. Furthermore, the SOS-PD scale contains one unique item that asks if family members do not recognize their child's behavior and an affirmative answer indicates pediatric delirium. Since development of novel tools requires training and time, leveraging the SOS-PD can facilitate rapid deployment to the bedside.

There are several assessment tools for pediatric delirium, including the PreSchool CAM-ICU (psCAM-ICU) and Cornell Assessment of Pediatric Delirium tools that are already developed and available in Japan. However, previous reports revealed that 54% of Japanese hospitals have introduced an iatrogenic withdrawal symptoms

assessment scale while only 21% of Japanese hospitals have introduced a pediatric delirium assessment scale [8]. Therefore, the gap between evaluation and analysis gap with respect to pediatric delirium remains wide in Japan, most likely because of the low adoption rates of existing instruments. One possible reason for this is that these scales do not account for the overlap of symptoms in pediatric delirium and iatrogenic withdrawal symptoms. A screening tool for early recognition of both iatrogenic withdrawal symptoms and pediatric delirium would be practical in clinical care and nurses can provide timely treatment. The aim of this study was thus to both translate the Japanese SOS-PD and perform a cross culture validation of the pediatric delirium dimension of the Japanese SOS-PD.

## Methods

### Study design

This study included two phases. The first phase was translation and second phase was validation. This study was a prospective cohort observational study and was conducted in a pediatric intensive care unit.

### Setting

We enrolled patients in our mixed-use (eight-bed post-surgical and internal medicine) pediatric ICU at the University of Tsukuba from October 2018 to January 2020.

### Participants

The original SOS-PD is validated from 3 months of age but, from our experience of translating and validating the Japanese versions of the psCAM-ICU [9] and CPAD [10], we considered that components of the SOS-PD were as valid for newborns as the Cornell Assessment of Pediatric Delirium scale. While no direct studies have been conducted on infants using the SOS-PD, our experience with the validation of the reliability and validity of the Japanese version of the Cornell Pediatric Delirium Assessment (CPAD) in newborns [10] led us to extend the inclusion criteria.

For this reason, we extended the inclusion criteria and included children from birth (newborns with a gestational of >40 weeks) up to 20 years old. Patients were excluded for the following reasons: neurological abnormalities (e.g., traumatic brain injury, encephalitis) or coma patients scoring under −4 on the Richmond Agitation–Sedation Scale (RASS) during assessment.

### Time course

Time courses were considered as standard round times, which might have occasionally changed based on the clinical situation. Tests were consistently administered during routine team rounds, which occurred every Monday at 1 PM. As a result, there was always a one-week interval between evaluations. Subsequent test timing was consistent, ensuring weekly assessments, although minor adjustments may have been made to accommodate urgent clinical needs. We evaluated every patient in the PICU during team rounds and reported all consecutive data in this study.

### Data collection

We extracted characteristics, including age, sex, presence of trisomy, developmental delay, mechanical ventilation status, and diagnosis for pediatric ICU admission from medical records. Datasets from newborn patients to 20 years old were analyzed as indicated in the Participants section. We extracted characteristics, including age, sex, presence of trisomy, developmental delay, mechanical ventilation status, and diagnosis, for pediatric ICU admission from medical records.

### Instrument – SOS-PD scale

The SOS-PD scale consists of 17 items that evaluate pediatric delirium symptoms. The presence of the items "hallucinations" or "parents do not recognize their child's behavior" are scored as 4 points while other items are scored as 1 point.

A total score of over 4 points indicates the presence of pediatric delirium. A previous multicenter study reported that the SOS-PD had an overall sensitivity of 92.3% and specificity of 96.5% compared to psychiatric diagnosis [11].

**Translation procedure**

We translated the SOS-PD from English into Japanese using forward/backward translation [12], a standardized process we used previously that maintains meanings between the original and translated languages [13,14]. We included eight medical staff members (two clinical nurses, two intensivists, two pediatricians, two researchers) and three translation staff (Japanese professional translator, an English bilingual speaker in public health, and a native English-speaking scientist) for this version. The translation process began with the principal investigator requesting the original author of the questionnaire for permission to translate. This initial step was crucial for ensuring that the translation would maintain the integrity of the original material. Following this, the principal investigator, along with the translators, received detailed information about the concept of the questionnaire. This foundational understanding enabled two independent translators (YM,HH) to carry out the translation from English to Japanese accurately.

The next phase involved reconciliation, where comparisons were made both with the original author's work and between the translations to ensure consistency and fidelity to the original text. To further verify the accuracy of the translation, a back translation was conducted. This involved translating the Japanese version back into English by a translator who was not familiar with the original English version, thereby providing an unbiased perspective on the translation's accuracy.

The original author then reviewed the back-translated manuscript, assisting the principal investigator in checking the accuracy of the translations. This step was followed by a harmonization process, where the original author and the principal investigator worked together to ensure that the translated version faithfully represented the original text.

To assess the comprehensibility of the translated manuscript, two parents were invited to evaluate it. Their feedback was crucial in identifying any points that were difficult to understand, although no issues requiring correction were found. Subsequent to this evaluation, the manuscript underwent test editing to correct any grammatical or typing errors, ensuring the final version was polished and readable. The process culminated in the creation of a final report, which included a detailed description of the methods used throughout the translation process. This comprehensive approach ensured that the translated questionnaire was both accurate and understandable to the target audience.

We used the STARD (Standards for Reporting Diagnostic accuracy studies) statement [15] for reference, as this study evaluated the diagnostic accuracy of the Japanese SOS-PD.

**Validation procedure**

**Reliability test.** To evaluate the reliability of the Japanese SOS-PD, we examined interobserver reliability between researchers (principal and ancillary) and evaluated reliability between psychiatrists (principal and ancillary).

**Criterion validation test.** A summary of the study flow is shown in Fig 1. We evaluated PD by pediatric intensivists and researchers simultaneously. Researchers, consisting of nurses with over 10 years of clinical experience in addition to research doctorates, evaluated PD by our Japanese translation of the SOS-PD while the pediatric intensivist assessed pediatric delirium according to the Vanderbilt Assessment for Delirium in Infants and Children (VADIC) [16] assessment sheet. However, clinical psychiatrists diagnosed pediatric delirium based on the chart review and VADIC. As VADIC is originally an assessment sheet for clinical psychiatrists, the pediatric intensivist's decision was only an assessment and verification and diagnosis based on VADIC results were actually conducted by a psychiatrist in accordance with the 5th edition of the Diagnostic and Statistical Manual of Mental Disorders (DSM-5) criteria [17].

To increase accuracy, we chose to assess PD by pediatric intensivist but diagnose PD by psychiatrist as, in Japan, psychiatrists do not frequently assess or gauge pediatric delirium severity as in other countries. The VADIC was previously reported [16] as a systematic assessment method for pediatric delirium. We previously used this adaptative method in a

 

## Flow of the research

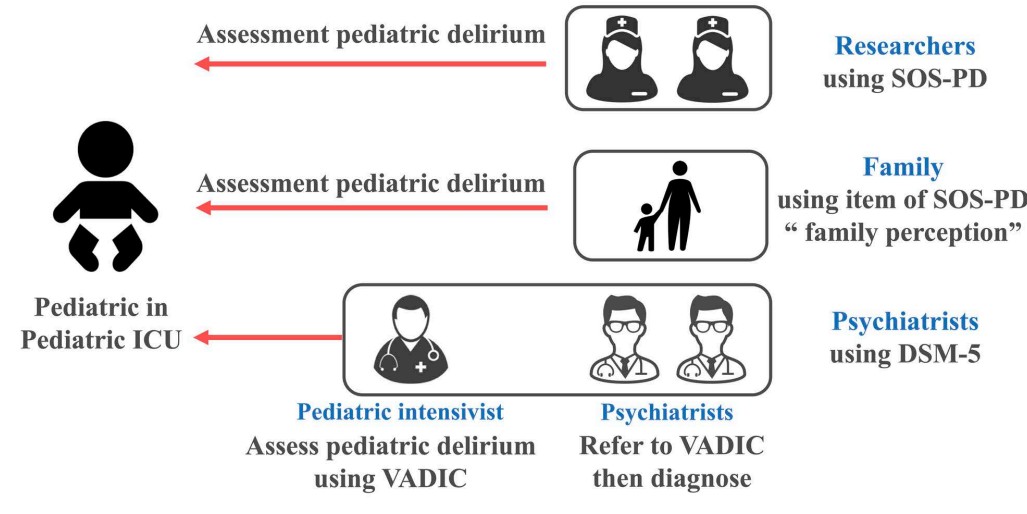

## Statistics

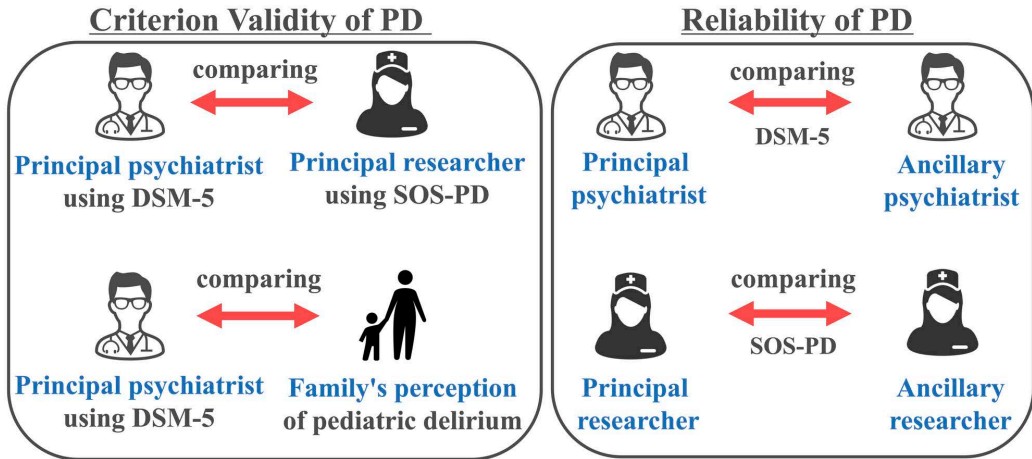

**Fig 1. Summary of this research.** DSM-5: 5th Diagnostic and Statistical Manual of Mental Disorders, SOS-PD: Sophia Observation withdrawal Symptoms and Delirium scale, VADIC: Vanderbilt Assessment for Delirium in Infants and Children.

validation study of the Japanese version of the psCAM-ICU [18], and also the Japanese version of the Cornell Assessment of Pediatric Delirium [10]. All diagnoses were done by principal psychiatrists because of their exceptional experience, with observed disagreement between principal and ancillary psychiatrists serving as the basis for reliability ratings as described above.

**Family perception of pediatric delirium.** We evaluated family perception of pediatric delirium using the SOS-PD question "Parents do not recognize their child's behavior" then compared diagnosed pediatric delirium by psychiatrists in accordance with DSM-5 criteria [17].

**Prevalence, positive rates and symptoms of pediatric delirium.** We show the prevalence of pediatric delirium based on the number of children and a positive rate of pediatric delirium based on the number of total observations. We

also show the positive rate of pediatric delirium symptoms in all observations of pediatric delirium as collected by our version of the SOS-PD.

**Statistical analysis.** For dichotomous variables, we used Fisher's exact test for two groups and t-testing for continuous variables in the univariate analysis. Criterion validity was determined by the accuracy of the pediatric delirium dimension of the SOS-PD evaluation compared with the psychiatric results [17] as measured by specificity, sensitivity, negative predictive value (NPV), positive predictive value (PPV), likelihood ratio for positive results (LR+), and likelihood ratio for negative results (LR-) with 95% confidence intervals (95%CI). These measures assess the degree to which the SOS-PD identifies pediatric delirium and also indicates sensitivity and specificity. We also evaluated reliability between researchers (principal and ancillary) and psychiatrists (principal and ancillary) by using the weighted Cohen's kappa coefficient [19] and intraclass correlation coefficient (ICC) for total score [20]. The weighted Cohen's kappa coefficient measures how much agreement there is between two or more raters when they classify items into categories. It considers the presence of agreement as well as its strength. For example, if both rate a patient's condition as "severe," the agreement is stronger than if one rates it as "mild" and the other as "severe." The weighted Cohen's kappa coefficient is widely used to evaluate the cross-classification of ordinal variables and ICC is also frequently used for repeated measures where multiple observations are collected from the same individual. This measure indicates the reliability of multiple iterations of the same test. For instance, if a doctor measures a patient's condition on different days, ICC shows how similar the results are. For this study, we selected ICC with a two-way random-effects model (2, k) [21]. This model helps to understand the consistency of measurements considering both differences between patients and raters. Furthermore, in case of repeated measurements, we conducted a sensitivity analysis for sensitivity and specificity on the first observation for each patient (i.e., the first assessment by the pediatric physician and the first observation of the pediatric delirium score) for all patients. We also evaluated the accuracy of family perception of pediatric delirium similarly and adequate cut-off points were determined by receiver operating characteristic (ROC) curve analysis, which evaluates SOS-PD performanc by plotting its sensitivity (true positives) against specificity (false positives) for different cut-off points. The ROC curve the reveals the trade-off between catching as many true cases as possible while avoiding false positives. We also evaluated criterion validation with and without mechanical ventilation status.

**Sample size calculation.** Sample size was calculated based on pediatric delirium. Based on our previous report, we assumed a specificity of 96.5% and estimated the prevalence of pediatric delirium as 50%. We determined a sample size based on the formula of Flahault et al [22] and 110 observations would be required for a significance level (α) of 0.05 and test power (1-β) of 0.80.

## Ethics

This study was carried out under laws equivalent to or derived from the principles of the Declaration of Helsinki and was approved by the University of Tsukuba Institutional Review Board (approval # H28-085). In addition, this study was conducted under the approval of the Ethics Committee and was performed using an opt-out format, where participants were given the opportunity to decline participation after being informed about the study.

## Results

### Translated Japanese version of SOS-PD

Our teams sent the SOS-PD scale through the back-translation cycle three times to minimize ambiguity then obtained approval from the original author as a final verification of accuracy. Major changes during the translations were in sentences such as "Parents do not recognize their child's behavior", "Parents perceive their child's behavior as very different or unrecognizable in comparison with what they are accustomed to when the child is ill or in hospital" and "this is not my child." It was difficult to maintain subtle nuances between languages since Japanese contains many phrases for recognition. Additionally, we added the Japanese mimetic (sound symbolic) words for "Tremors" and "Muscle tension" sections

to make it easier to understand for Japanese users. All other symptoms and the SOS-PD instructions were translated relatively fluently and no other problems were encountered. (Japanese version of SOS-PD is freely available from online repository in University of Tsukuba [23])

## Demographic data

From October 2018 to January 2020, we evaluated a total of 125 observations with a median of 1 (IQR 1–2) observation per patient in 67 patients for this study. We recorded 255 initial observations but excluded 130 due to the following criteria: 52 were in coma patients, 73 were related to brain dysfunction and 5 were in those aged over 20 years old or GA<40 weeks at observation time.

Table 1 shows the demographic data of participating patients. Patients were a median of 15 (IQR 5–54) months of age, with females accounting for 61%. Within all included patients, 10 assessments were evaluated in children under 3 months of age with 70% of these diagnosed as cardiac surgical cases. Within the total patient pool, 19% had Down syndrome and 32% patients had developmental delays. Approximately 33% of patients were evaluated during mechanical ventilation and 67% of patients were evaluated without mechanical ventilation.

## Prevalence, positive rates and symptoms of pediatric delirium

We show the prevalence of pediatric delirium in participating patients in Table 1. A total of 24 patients out of 67 experienced pediatric delirium (36%) and approximately 15% (n=10) of the mechanical ventilation patients were diagnosed with pediatric delirium while 21% (n=14) of the non-ventilated patients were diagnosed with pediatric delirium. The

**Table 1. Baseline characteristics and research evaluation units.**

| Variable | Participating Patients N=67 |
|---|---|
| Months, (IQR) | 15 (5, 54) |
| Age categories, n (%) | |
| 0–3 months | 10 (15) |
| 3–12 months | 18 (27) |
| 1–5 years | 26 (39) |
| 6–18 years | 13 (19) |
| Female, n (%) | 41 (61) |
| Diagnoses | |
| Cardiac Surgical, n (%) | 47 (70) |
| Medical, n (%) | 14 (20) |
| Abdominal Surgical, n (%) | 5 (7) |
| Thoracic Surgical, n (%) | 1 (1) |
| Trisomy Patients, n (%) | 9 (13) |
| Developmental Delay, n (%) | 19 (28) |
| Mechanical Ventilation [a] | 22 (33) |
| Prevalence of Delirium, n (%) | |
| Total, n (%) | 24 (36) |
| With Mechanical Ventilation, n (Within the Group, %) | 10 (15) |
| Without Mechanical Ventilation, n (Within the Group, %) | 14 (21) |

SD=standard deviation.

positive rate of pediatric delirium in total assessments (n = 125) was 30% (n = 38) and, within this group, 37% (n = 13) of the mechanical ventilation observations were diagnosed with pediatric delirium while 27% (n = 25) of the non-ventilation observations were pediatric delirium diagnoses. We show the positive rate of the pediatric delirium symptoms in all observations of pediatric delirium as judged by the SOS-PD in Fig 2. Diagnosed pediatric delirium patients had multiple and more pediatric delirium symptoms compared with the total observation data. Of all the assessments, the main symptoms were attentiveness (89%), fluctuating (89%), and lack of eye contact (86%) while acute onset of symptoms (86%) was frequently observed in diagnosed pediatric delirium observations.

## Reliability of the Japanese SOS-PD

The Japanese SOS-PD showed high reliability between the researchers (κ = 0.95, 95%CI [0.93–0.98], ICC = 0.98, 95%CI [0.97–0.98]) (Table 2). This trend was also observed in mechanical ventilation patients (κ = 0.96, 95%CI [0.92–1.0], ICC = 0.98, 95%CI [0.97–0.99]) and non-ventilated patients (κ = 0.94, 95%CI [0.90–0.98], ICC = 0.97, 95%CI [0.96–0.98]). Reliability for the psychiatric evaluations with DSM-5 criteria was high (κ = 0.92, 95%CI [0.85–0.99], ICC = 0.96, 95%CI [0.95–0.97]) and this trend was also observed in mechanical ventilation patients (κ = 0.93, 95%CI [0.81–1.00], ICC = 0.97, 95%CI [0.95–0.98]) and non-ventilated patients (κ = 0.91, 95%CI [0.82–1.00], ICC = 0.96, 95%CI [0.94–0.97]). We also evaluated the consistency of the diagnoses between the psychiatrist and pediatric intensivist and the reliability of the intensivist was high in total observations (κ = 0.90, 95%CI [0.82–0.98], ICC = 0.95, 95%CI [0.94–0.96]), a trend also observed in mechanical ventilation patients (κ = 0.81, 95%CI [0.61–1.00], ICC = 0.90, 95%CI [0.83–0.95]) and non-ventilated patients (κ = 0.94, 95%CI [0.87–1.00], ICC = 0.97, 95%CI [0.96–0.98]).

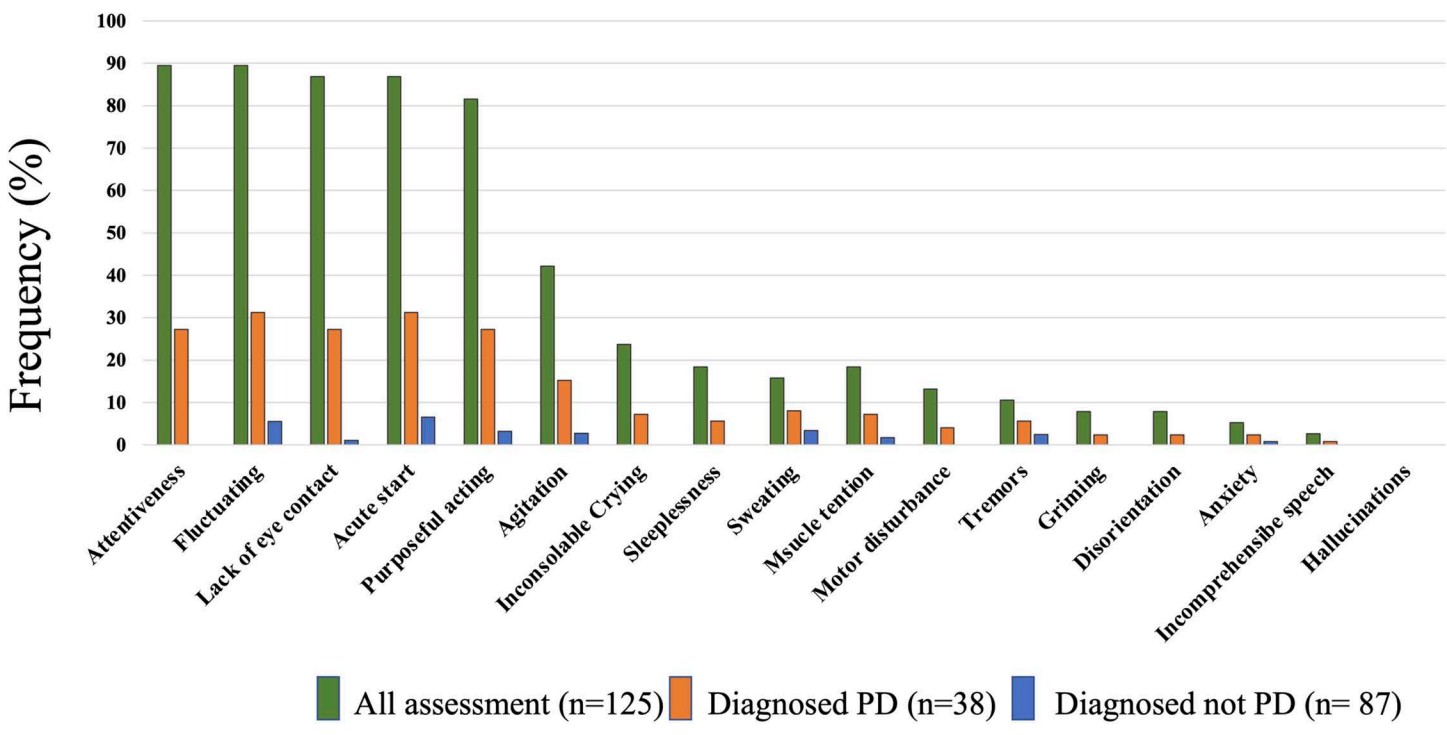

**Fig 2. Symptoms of the pediatric delirium scale.** This figure shows the prevalence of each symptoms of the pediatric delirium scale.

**Table 2. Reliability of the Japanese SOS-PD.**

| Variable | Without Mechanical Ventilation N = 90 | With Mechanical Ventilation N = 35 | Total Observations N = 125 |
|---|---|---|---|
| **Reliability of SOS-PD** | | | |
| Weighted kappa[a] | 0.94 (0.90 - 0.98) | 0.96 (0.92 - 1.0) | 0.95 (0.93 - 0.98) |
| ICC[b] | 0.97 (0.96 - 0.98) | 0.98 (0.97 - 0.99) | 0.98 (0.97 - 0.98) |
| **Reliability of Psychiatric Diagnoses** | | | |
| kappa[a] | 0.91 (0.82-1.00) | 0.93 (0.81-1.00) | 0.92 (0.85-0.99) |
| ICC[b] | 0.96 (0.94-0.97) | 0.97 (0.95-0.98) | 0.96 (0.95-0.97) |

Data are values (95% confidence interval).

[a]Data are kappa coefficients (95% confidence interval).

[b]Data are intraclass correlation coefficients (95% confidence interval).

For the subgroup analysis, Table 3 shows the reliability of the Japanese version of the SOS-PD for children under 3 months of age, based on a total of 22 observations. The weighted kappa for SOS-PD is 0.96, with a 95% confidence interval ranging from 0.92 to 1.00, and the ICC for SOS-PD is also 0.96, with a 95% confidence interval of 0.91 to 0.98.

Regarding the reliability of psychiatric diagnoses, the kappa coefficient and ICC both achieved 1.00, showing perfect agreement between the psychiatrists' diagnoses.

## Criterion validation of pediatric delirium

We performed 125 observations on 67 patients. We analyzed adequate cut-off points by ROC curve and 4 was the most appropriate cut-off point. Fig 3 shows the criterion validity of the Japanese SOS-PD using a cut-off point of 4 compared to psychiatric diagnoses using DSM-5 criteria. We also performed a sensitivity analysis, using the first assessment in each patient (n = 67) (see Supplemental File 1). The sensitivity and specificity of the Japanese version of SOS-PD was similar to the result of total observations (sensitivity: 0.9, 95% CI: 0.76–1.00, specificity: 1.00, 95% CI: 1.00–1.00) and the ROC curve suggested point 4 was the most appropriate cut-off point as the result of total observation.

Based on this cut off point, the Japanese version of SOS-PD shows a sensitivity of 0.92 (95% CI: 0.84–1.00), specificity of 0.97 (95% CI: 0.94–1.00), PPV of 0.94 (95% CI: 0.86–0.98), NPV of 0.96 (95% CI: 0.93–0.98), LR+ of 40 (95% CI: 14.3–125.4), and LR- of 0.08 (95% CI: 0.04–0.17) within 125 total observations (Table 4). In patients with mechanical ventilation, the Japanese SOS-PD also had high sensitivity (0.92, 95% CI: 0.77–0.92) and high specificity. (1.00, 95%

**Table 3. Reliability of the Japanese SOS-PD under 3 months.**

| Variable | Total Observations N = 22 |
|---|---|
| **Reliability of SOS-PD** | |
| Weighted kappa[a] | 0.96 (0.92 - 1.00) |
| ICC[b] | 0.96 (0.91 - 0.98) |
| **Reliability of Psychiatric Diagnoses** | |
| kappa[a] | 1 (1.00-1.00) |
| ICC[b] | 1 (1.00-1.00) |

Data are values (95% confidence interval).

[a]Data are kappa coefficients (95% confidence interval).

[b]Data are intraclass correlation coefficients (95% confidence interval).

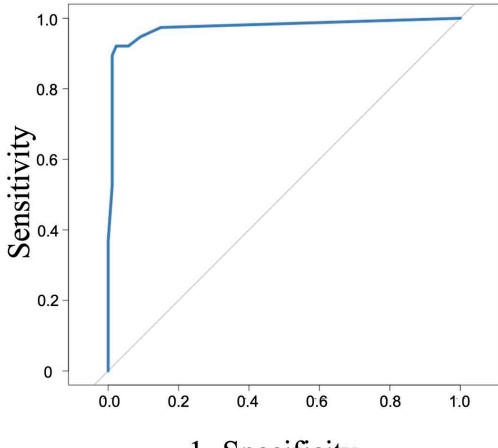

| Cut-off point of SOS-PD | Sensitivity (95%CI) | | Specificity (95%CI) | |
|---|---|---|---|---|
| 1 | 0.97 | (0.91-0.97) | 0.85 | (0.77-0.91) |
| 2 | 0.94 | (0.86-1.00) | 0.9 | (0.83-0.96) |
| 3 | 0.92 | (0.81-1.00) | 0.94 | (0.88-0.98) |
| 4 | 0.92 | (0.84-1.00) | 0.97 | (0.94-1.00) |
| 5 | 0.89 | (0.78-0.97) | 0.98 | (0.96-1.00) |
| 6 | 0.52 | (0.36-0.68) | 0.98 | (0.96-1.00) |
| 7 | 0.36 | (0.23-0.52) | 1.00 | (1.00-1.00) |
| 8 | 0.28 | (0.15-0.44) | 1.00 | (1.00-1.00) |
| 9 | 0.15 | (0.05-0.28) | 1.00 | (1.00-1.00) |
| 10 | 0.05 | (0-0.13) | 1.00 | (1.00-1.00) |

Data are shown as value (95% confidence interval).
confidence interval was obtained by boot strapping method.

**Fig 3. ROC curve of SOS-PD.** We analyzed adequate cut-off points by ROC curve and 4 was the most appropriate cut-off point.

**Table 4. Criterion validity of the Japanese SOS-PD.**

| Variable | Without Mechanical Ventilation N = 90 | With Mechanical Ventilation N = 35 | Total Observations N = 125 |
|---|---|---|---|
| **Validity of SOS-PD** | | | |
| Sensitivity | 0.88 (0.76 - 0.93) | 0.92 (0.77 - 0.92) | 0.92 (0.84 - 1.00) |
| Specificity | 0.96 (0.92 - 0.99) | 1.00 (0.91 - 1.00) | 0.97 (0.94 - 1.00) |
| PPV | 0.91 (0.79 - 0.97) | 1.00 (0.83 - 1.00) | 0.94 (0.86 - 0.98) |
| NPV | 0.95 (0.91 - 0.97) | 0.95 (0.87 - 0.95) | 0.96 (0.93 - 0.98) |
| LR+ | 28.6 (9.98 - 91.05) | ∞ (8.64 - ∞) | 40 (14.3 - 125.4) |
| LR- | 0.12 (0.06 - 0.25) | 0.07 (0.07 - 0.25) | 0.08 (0.04 - 0.17) |

PPV: positive predictive values NPV: negative predictive values

LR+: likelihood ratio for positive results LR-: likelihood ratio for negative results

Data are values (95% confidence interval)

a: Data are kappa coefficients (95% confidence interval)

b: Data are intraclass correlation coefficients (95% confidence interval)

CI: 0.91–1.00). We saw the same trend in non-ventilated patients (sensitivity: 0.88, 95% CI: 0.76–0.93, specificity: 0.96, 95% CI: 0.92–0.99).

In Table 5, A subgroup analysis of the validity of the Japanese version of the SOS-PD for children under 3 months of age is presented, based on a total of 22 observations. The Japanese version of the SOS-PD demonstrates a sensitivity of 0.66 (95% CI: 0.09–0.99), specificity of 1.00 (95% CI: 0.82–1.00), PPV of 1.00 (95% CI: 0.15–1.00), NPV of 0.95 (95% CI: 0.79–0.98), and LR- of 0.33 (95% CI: 0.07–1.65).

## Criterion validation of pediatric delirium assessments by family

We show the accuracy of family perception of pediatric delirium compared to psychiatric diagnosis using DSM-5 criteria in Table 6. Family perceptions of pediatric delirium showed a sensitivity of 0.91 (95% CI: 0.72–0.98), specificity of 0.91 (95%

**Table 5. Validation of the Japanese SOS-PD in infants under 3 months.**

| Variable | Total Observations N = 22 |
|---|---|
| **Validity of SOS-PD** | |
| Sensitivity | 0.66 (0.09 - 0.99) |
| Specificity | 1.00 (0.82 - 1.00) |
| PPV | 1.00 (0.15 - 1.00) |
| NPV | 0.95 (0.79 - 0.98) |
| LR+ | – |
| LR- | 0.33 (0.07 - 1.65) |

PPV: positive predictive values NPV: negative predictive values.

LR+: likelihood ratio for positive results LR-: likelihood ratio for negative results.

Data are values (95% confidence interval).

a: Data are kappa coefficients (95% confidence interval).

b: Data are intraclass correlation coefficients (95% confidence interval).

**Table 6. Criterion validity of family perception of pediatric delirium.**

| Variable | Without Mechanical Ventilation N = 20 | With Mechanical Ventilation N = 16 | Total Observations N = 36 |
|---|---|---|---|
| Sensitivity | 0.83 (0.55 - 0.83) | 1.0 (0.70 - 1.00) | 0.91 (0.72-0.98) |
| Specificity | 1.00 (0.87 - 1.00) | 0.80 (0.62 - 0.80) | 0.91 (0.81-0.95) |
| PPV | 1.00 (0.66 −1.00) | 0.75 (0.30-0.75) | 0.84 (0.66-0.90) |
| NPV | 0.93 (0.82- 0.93) | 1.00 (0.77- 1.00) | 0.95 (0.85-0.99) |
| LR+ | – | 5.0 (1.87-5.00) | 11.0 (3.9-19.6) |
| LR- | 0.16 (0.16- 0.51) | 0.00 (0.00- 0.47) | 0.09 (0.01-0.34) |

PPV: positive predictive values.

NPV: negative predictive values.

LR+: likelihood ratio for positive results.

LR-: likelihood ratio for negative results.

Data are values [95% confidence interval].

CI: 0.81–0.95), PPV of 0.84 (95% CI: 0.66–0.90), NPV of 0.95 (95% CI: 0.85–0.99), LR- of 0.09 (95% CI: 0.01–0.34) and LR+ of 11.0 (95% CI: 3.9–19.6) within 36 total observations. In patients with mechanical ventilation, family perceptions of pediatric delirium had high sensitivity (1.0, 95% CI: 0.7–1.0) and relatively high specificity (0.8, 95% CI: 0.62–0.8) while relatively high specificity compared with sensitivity was observed in non-ventilated patients (sensitivity: 0.83, 95% CI: 0.55–0.83, specificity: 1.0, 95% CI: 0.87–1.0).

## Discussion

In this study, we developed a Japanese Version of the SOS-PD scale via the rigorous back-translation method that demonstrates high criterion validity and reliability. Additionally, this is the first study to use the Japanese SOS-PD to verify a high criterion validity of family perceptions of pediatric delirium.

We evaluated adequate cut-off points of our SOS-PD translation and a cut-off point of 4 showed high sensitivity and specificity. This cut-off point and the criterion validity were the same as the original SOS-PD [11]. As a previous study

evaluated 485 assessments in a multicenter study [11], the accuracy of original SOS-PD was validated and our Japanese version is in line with these results. Additionally, the criterion validity was and reliability of pediatric delirium testing did not differ between mechanical ventilation and non-ventilation patients, indicating that the Japanese SOS-PD is useful for both situations with as high a criterion validity and reliability as the original instrument.

About the symptoms of delirium and withdrawal, Attentiveness (89%), fluctuating (89%), lack of eye contact (86%), and acute stat (86%) were frequently observed in our diagnosed pediatric delirium observations. These four symptoms are key components of the criteria for delirium in the DSM-5 and, thus, our results are reasonable.

Regarding the inclusion of children under 3 months of age, we expanded the scope of the SOS-PD by including this subgroup, which was not part of the original study. The decision was based on our prior experience with pediatric delirium assessment tools and the applicability of the SOS-PD to younger infants. Our analysis showed that the Japanese SOS-PD remained highly reliable for this group. While this age group presents unique challenges due to developmental differences, the SOS-PD was effective in identifying pediatric delirium even in infants under 3 months of age. This finding highlights the potential of the SOS-PD to be used across a broader age range in pediatric delirium detection.

Regarding the study population, while this study primarily involved cardiac patients and many participants were under 12 months of age, our findings must be considered in the context of the high variability of pediatric ICU populations. The PICU is a diverse environment that treats patients with different underlying conditions and, in our hospital, cardiac surgery patients are the primary population, which is similar to PICUs worldwide that treat a significant number of post-cardiac surgery patients. However, in PICUs that focus on trauma or other specialties, the prevalence of delirium may differ. Therefore, further research is needed to explore the applicability of the Japanese SOS-PD across various PICU populations. Comparisons with studies in other types of PICUs will help establish the broader generalizability of these findings.

Furthermore, while this study utilized experienced researchers to evaluate the SOS-PD, future research should validate the scale with bedside nurses in real-world PICU settings. Nurses, who have the most frequent contact with patients, are in the best position to notice subtle changes in patient conditions. Therefore, involving nurses in evaluating the SOS-PD is essential to better understand its utility in daily clinical practice. In addition, education on how to effectively use the scale in clinical settings is crucial, and training programs should be implemented to ensure that nurses can properly apply the scale in their daily routines. This will allow for clearer insight into how the SOS-PD can be integrated into routine clinical care.

Our study also showed that parents were able to diagnose PD quickly and with only one component of the SOS-PD ("parents do not recognize their child's behavior") while medical workers needed 17 components of the SOS-PD. The fact that pediatric evaluation is hindered by a lack of linguistic contact is the main reason why pediatric delirium cannot be assessed the same as adult delirium. Tools, such as the Preschool Confusion Assessment Method for the ICU (psCAM-ICU) [24], Cornell Assessment of Pediatric Delirium [25], and SOS-PD [7] exist but no physician or nurse can know an individual patient better than the parents. In our data, family perceptions of pediatric delirium had 100% sensitivity for pediatric delirium in mechanical ventilation patients and 100% specificity for pediatric delirium in non-ventilated patients. These findings support the inclusion of family assessments in the clinical setting, as suggested by studies showing that family members can effectively identify signs of delirium in critically ill children [26].

While this study highlights the high accuracy of family assessments, it also suggests important implications for healthcare systems. As a previous study described [27], integrating family assessments into clinical practice could provide an additional layer of observation, particularly in settings where healthcare staff may have limited time with each patient. Structured tools like SOS-PD could be used to guide family members in identifying changes in their child's behavior, enabling earlier detection and intervention for pediatric delirium.

However, there are potential risks and limitations to relying on family observations. Family members may lack clinical expertise and could misinterpret symptoms, confusing them with other factors such as pain, sedation, or recovery from surgery. Therefore, family assessments should complement clinical evaluations by healthcare professionals, not replace

them as a previous study stated [28]. Moreover, in some ICU settings, family access may be restricted due to infection control or other reasons, which could limit the feasibility of using family assessments. In these cases, alternative methods such as phone consultations or digital communication tools might help involve families in the assessment process.

Future studies will hopefully delineate the risk factor and related outcome of pediatric delirium, and we recommend the regular use of SOS-PD. this tool in the PICU at discharge to gather immediate feedback. Crucially, the feedback should directly inform and lead to tangible changes in PICU practices. This could involve structured training for staff on how to analyze feedback and apply it to improve areas such as patient communication and care protocols.

## Limitations

There are some limitations for this study. As most participating patients were cardiac patients and 40% of the observations were examined for 3–12 months, this might reduce the generalizability of the results for the general medical-surgical PICU setting. Additionally, our evaluation of patients under 3 months of age runs contrary to the original study and 10 assessments in children under 3 months of age were included in this study. This cohort might reduce the accuracy of our Japanese version of the SOS-PD. Moreover, repeated measurements in this study might have introduced bias in prediction of the occurrence of pediatric delirium. In our study we evaluated SOS-PD by researchers rather than bedside nurses, which might reduce the generality for actual ICU conditions since the Japanese version of the SOS-PD still needs to be evaluated by bedside nurses. However, we additionally performed a sensitivity analysis using the first assessment in each patient and the result was similar to the results of the total observation. Therefore, the bias of repeated measurements did not affect the result. Delirium is highly variable and must be repeatedly evaluated. Thus, we believe each observation was independent and repeated measurements are more clinically relevant. Finally, we acknowledge that a pediatric psychiatrist is the gold standard for pediatric delirium diagnosis but, in Japan, there is no pediatric specialty for psychiatry so we used our intensivist-psychiatrist combination to arrive at pediatric delirium diagnoses. Although this is a workaround that could impart bias to the results, we did blind the psychiatrist to the intensivist diagnoses and reported an excellent correlation between the 2 judgements.

## Conclusion

We translated and conducted a validation study of the Japanese version of the SOS-PD and showed an accuracy as high as the original SOS-PD. Moreover, we found that family perception of pediatric delirium is surprisingly accurate and may be used as a resource in an family-centered care -integrated clinical setting.

## Author contributions

**Conceptualization:** Yoshiaki Inoue, Yujiro Matsuishi, Erwin Ista.

**Data curation:** Haruhiko Hoshino.

**Formal analysis:** Yujiro Matsuishi.

**Funding acquisition:** Yujiro Matsuishi, Yoshiaki Inoue.

**Investigation:** Haruhiko Hoshino, Yuki Enomoto, Takahiro Kido, Asaki Matsuzaki, Joshua Gallagher.

**Methodology:** Yujiro Matsuishi, Asaki Matsuzaki.

**Project administration:** Nobutake Shimojo.

**Resources:** Haruhiko Hoshino, Yuki Enomoto, Takahiro Kido, Asaki Matsuzaki.

**Supervision:** Yoshiaki Inoue.

**Writing – original draft:** Yujiro Matsuishi.

**Writing – review & editing:** Bryan J. Mathis.

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
