## [Decision Letter · Decision Letter 0]

23 Jan 2025

PONE-D-24-28949Verifying the accuracy of Japanese version of the pediatric delirium assessment scale: SOS-PD and the high accuracy of family assessments of pediatric deliriumPLOS ONE

Dear Dr. Matsuishi,

Thank you for submitting your manuscript to PLOS ONE. After careful consideration, we feel that it has merit but does not fully meet PLOS ONE’s publication criteria as it currently stands. Therefore, we invite you to submit a revised version of the manuscript that addresses the points raised during the review process.

We look forward to receiving your revised manuscript.

Kind regards,

Sidra Kaleem Jafri

Academic Editor

PLOS ONE

Reviewers' comments:

Reviewer's Responses to Questions

**Comments to the Author**

1. Is the manuscript technically sound, and do the data support the conclusions?

Reviewer #1: Yes

Reviewer #2: Yes

2. Has the statistical analysis been performed appropriately and rigorously? 

Reviewer #1: Yes

Reviewer #2: Yes

3. Have the authors made all data underlying the findings in their manuscript fully available?

Reviewer #1: Yes

Reviewer #2: Yes

4. Is the manuscript presented in an intelligible fashion and written in standard English?

Reviewer #1: Yes

Reviewer #2: Yes

5. Review Comments to the Author

Reviewer #1: Dear authors here are some recommendations for improving the manuscript:

- Provide a more detailed analysis or discussion regarding the inclusion of children under 3 months of age, as this group was not part of the original study. Explain how this age group might influence the results, and if possible, provide separate analyses or interpretations for this subgroup.

- While the study highlights the high accuracy of family assessments, consider expanding the discussion on the implications of this finding. For example, how might healthcare systems integrate family assessments in clinical practice? Are there any potential drawbacks or risks associated with relying on family observations in different ICU settings?

- Make the data availability statement clearer by indicating how interested researchers can request access to the data for future studies. Even if the data are restricted, providing a clear path for legitimate academic requests would enhance transparency and reproducibility.

- Since the study mainly involved cardiac patients and many participants were under 12 months old, consider providing a more detailed discussion about the generalizability of the findings to other pediatric ICU populations. If possible, compare your findings with studies conducted in other patient populations.

- Since the current study used experienced researchers to evaluate the SOS-PD, further validation by bedside nurses in a real-world PICU setting would be beneficial. You could recommend that future studies aim to evaluate the SOS-PD with the typical clinical staff who will be using the scale daily.

- While the statistical methods were appropriate, it might help to add a brief explanation or reference for certain techniques like the ROC curve analysis and the ICC for readers who may not be familiar with these terms. This would make the paper more accessible to a wider audience.

- Recommend that healthcare institutions consider providing structured training programs for the use of the SOS-PD scale, especially in translating research findings into everyday clinical practice. Highlight the importance of training both healthcare professionals and family members to enhance early detection of pediatric delirium.

Reviewer #2: Overall good work, however following revisions are suggested:

Page 7: Study design: Please mention the type of observational study.

Page 7: Participants: You mentioned that you have used the scale with infants based on experience, please cite any studies that have been done on infants in the past.

Page 8: Time Course: mention the duration between each test such as the first test was administered on the first day of admission, the second test was administered after a few hours etc.

Page 9: Translation Procedure: make a flow chart for the procedure so that it is easier for the reader to understand.

There aren’t enough references in the “Discussion” section. Please also add material related to the scoring with infants.

6. PLOS authors have the option to publish the peer review history of their article (what does this mean? ). If published, this will include your full peer review and any attached files.

**Do you want your identity to be public for this peer review?** For information about this choice, including consent withdrawal, please see our Privacy Policy .

Reviewer #1: **Yes: ** Ahmed Alhatemi

Reviewer #2: **Yes: ** Vardah Bharuchi

---

## [Author Response · Author response to Decision Letter 1]

26 Feb 2025

#1: Thank you very much for your thoughtful and constructive feedback. Your suggestions have been invaluable in strengthening the manuscript. We greatly appreciate your attention to detail and the depth of analysis you have provided. Below, we have outlined how we have addressed each of your recommendations.

Reviewer #1: Dear authors here are some recommendations for improving the manuscript:

- Provide a more detailed analysis or discussion regarding the inclusion of children under 3 months of age, as this group was not part of the original study. Explain how this age group might influence the results, and if possible, provide separate analyses or interpretations for this subgroup.

Answer: Thank you for your feedback. In response, we have explained the inclusion of children under 3 months of age in our study. The decision was based on our experience with other pediatric delirium tools and the applicability of SOS-PD to this younger population. The Japanese version of SOS-PD showed high reliability in this subgroup, with a weighted kappa and ICC of 0.96, supporting its validity for infants. Although including this age group presents challenges due to developmental differences, we found the scale effective in identifying pediatric delirium. We hope this analysis addresses your concerns.

- While the study highlights the high accuracy of family assessments, consider expanding the discussion on the implications of this finding. For example, how might healthcare systems integrate family assessments in clinical practice? Are there any potential drawbacks or risks associated with relying on family observations in different ICU settings?

Answer:We agree that family assessments have great potential for improving pediatric delirium detection. In response, we expanded the discussion on integrating family assessments into clinical practice, suggesting that structured tools like SOS-PD can guide families in identifying behavior changes for earlier delirium detection. However, we acknowledge potential risks, such as misinterpretation by family members without clinical expertise. Therefore, family assessments should complement, not replace, clinical evaluations. Additionally, in some ICU settings where family access may be restricted, we suggest exploring alternative methods, like phone consultations or digital tools, to involve families in assessments.

- Make the data availability statement clearer by indicating how interested researchers can request access to the data for future studies. Even if the data are restricted, providing a clear path for legitimate academic requests would enhance transparency and reproducibility.

Answer: Thank you for your suggestion. In response, we have updated the data availability statement to specify that researchers interested in accessing the data for future studies may contact Dr. Yujiro Matsuishi, the co-corresponding author. This ensures a clear path for academic requests and enhances transparency.

- Since the study mainly involved cardiac patients and many participants were under 12 months old, consider providing a more detailed discussion about the generalizability of the findings to other pediatric ICU populations. If possible, compare your findings with studies conducted in other patient populations.

Answer: Thank you for your valuable comment. We have expanded the discussion on the generalizability of our findings to other pediatric ICU populations. While this study primarily involved cardiac patients, and many participants were under 12 months of age, we emphasize that the findings should be considered within the context of various pediatric ICU populations. The PICU is a diverse environment that treats patients with different underlying conditions, and in our hospital, cardiac surgery patients are the primary population. This is similar to PICUs worldwide that treat a significant number of post-cardiac surgery patients. However, in PICUs that focus on trauma or other specialties, the prevalence of delirium and its underlying causes may differ.

As such, further research is necessary to explore the applicability of the Japanese SOS-PD across various PICU populations. Comparing our findings with studies in other types of PICUs, such as those focusing on trauma, neurological conditions, or respiratory diseases, will help establish the broader generalizability of these findings and validate the effectiveness of the Japanese SOS-PD in diverse clinical settings.

We hope this addition addresses your comment thoroughly and provides a clearer understanding of the broader implications of our findings.

- Since the current study used experienced researchers to evaluate the SOS-PD, further validation by bedside nurses in a real-world PICU setting would be beneficial. You could recommend that future studies aim to evaluate the SOS-PD with the typical clinical staff who will be using the scale daily.

Answer: Thank you for your valuable suggestion. We agree that structured training programs for the use of the SOS-PD scale are essential. We recommend that healthcare institutions implement training for both healthcare professionals and family members to ensure consistent use of the scale. Training healthcare staff, particularly nurses and physicians, will help them accurately identify pediatric delirium and intervene early. Additionally, educating family members to recognize signs of delirium can further enhance early detection.

We believe such programs will bridge the gap between research and clinical practice, improving outcomes for pediatric patients.

- While the statistical methods were appropriate, it might help to add a brief explanation or reference for certain techniques like the ROC curve analysis and the ICC for readers who may not be familiar with these terms. This would make the paper more accessible to a wider audience.

Answer: Thank you for your suggestion. In response, we have added brief explanations of the ROC curve analysis and the intraclass correlation coefficient (ICC) to make the paper more accessible to a wider audience. The ROC curve helps evaluate a test's diagnostic performance by comparing sensitivity against 1-specificity at different cut-off points, while ICC measures the consistency of repeated measurements. We believe these additions will clarify the statistical methods for readers who may not be familiar with these terms.

- Recommend that healthcare institutions consider providing structured training programs for the use of the SOS-PD scale, especially in translating research findings into everyday clinical practice. Highlight the importance of training both healthcare professionals and family members to enhance early detection of pediatric delirium

Answer: Thank you for your insightful suggestion. We agree that structured training programs for the use of the SOS-PD scale are essential, particularly when translating research findings into everyday clinical practice. In response to your comment, we recommend that healthcare institutions consider implementing comprehensive training programs for both healthcare professionals and family members.

We hope this addresses your suggestion and thank you for your thoughtful input.

Reviewer #2: Thank you very much for your thoughtful and constructive feedback. We greatly appreciate your valuable suggestions, which have significantly helped improve the manuscript. Below, we have outlined how we have addressed each of your comments.

Reviewer #2: Overall good work, however following revisions are suggested:

Page 7: Study design: Please mention the type of observational study.

Answer: Thank you for your suggestion. In response, we have clarified that the study is a prospective cohort observational study in the revised manuscript.

Page 7: Participants: You mentioned that you have used the scale with infants based on experience, please cite any studies that have been done on infants in the past.

Answer: Thank you for your comment. In response, we clarified that although no direct studies on infants using the SOS-PD exist, our experience with the Japanese version of the Cornell Pediatric Delirium Assessment (CPAD) in newborns, where we validated its reliability and validity, led us to extend the inclusion criteria. We will also reference relevant studies to support our approach.

Page 8: Time Course: mention the duration between each test such as the first test was administered on the first day of admission, the second test was administered after a few hours etc.

Answer: Thank you for your helpful comment. In response, we have revised the description of the time course to clarify that tests were consistently administered during routine team rounds, which occurred every Monday at 1 PM. As a result, there was always a one-week interval between evaluations, ensuring consistent timing between assessments. This revision emphasizes the regularity of the evaluations and the weekly interval between tests.

We hope this addresses your comment and provides the clarity you were seeking.

Page 9: Translation Procedure: make a flow chart for the procedure so that it is easier for the reader to understand.

There aren’t enough references in the “Discussion” section. Please also add material related to the scoring with infants.

Answer: In response to your comment regarding the translation procedure, we have created a flow chart to visually represent the steps involved in translating SOS-PD from English (E) to Japanese (J). This flow chart should make it easier for readers to understand the process and follow the steps taken during the translation.

Additionally, we have increased the number of references in the "Discussion" section to strengthen the arguments and provide additional context. We also addressed the use of the scale with infants, citing relevant studies to support the validity and reliability of the scale in this age group.

We hope these revisions meet your expectations and enhance the clarity and depth of the manuscript.

---

## [Editor Report · Decision Letter 1]

31 Mar 2025

Verifying the accuracy of Japanese version of the pediatric delirium assessment scale: SOS-PD and the high accuracy of family assessments of pediatric delirium

PONE-D-24-28949R1

Dear Dr. Matsuishi,

We’re pleased to inform you that your manuscript has been judged scientifically suitable for publication and will be formally accepted for publication once it meets all outstanding technical requirements.

Kind regards,

Sidra Kaleem Jafri

Academic Editor

PLOS ONE
---

## [Editor Report · Acceptance letter]

PONE-D-24-28949R1

PLOS ONE

Dear Dr. Inoue,

I'm pleased to inform you that your manuscript has been deemed suitable for publication in PLOS ONE. Congratulations! Your manuscript is now being handed over to our production team.

Kind regards,

on behalf of

Dr. Sidra Kaleem Jafri

Academic Editor

PLOS ONE